# Developing New Immunotherapy Approaches for Colorectal Cancer

**DOI:** 10.3390/cancers17243929

**Published:** 2025-12-09

**Authors:** Gregory Kelly, Bianca Nowlan, Simon Manuel Tria, Afshin Nikkhoo, Catherine Bond, Vicki Whitehall

**Affiliations:** 1Conjoint Gastroenterology Laboratory, QIMR Berghofer Medical Research Institute, Herston Rd, Herston, QLD 4006, Australia; greg.kelly@qimrb.edu.au (G.K.); bianca.nowlan@qimrb.edu.au (B.N.); simon.tria@qimrb.edu.au (S.M.T.); afshin.nikkhoo@qimrb.edu.au (A.N.); catherine.bond@qimrb.edu.au (C.B.); 2Faculty of Medicine, The University of Queensland, Brisbane, QLD 4102, Australia; 3Conjoint Internal Medicine Laboratory, Queensland Health, Brisbane, QLD 4006, Australia

**Keywords:** colorectal cancer, immunotherapy, CRISPR/Cas9 screens, in vivo models, co-culture models, organoids

## Abstract

Colorectal cancer (CRC) is the second leading cause of cancer-related deaths worldwide. While immunotherapies have shown promise in a subset of patients, they are largely ineffective for CRC. This limited sensitivity is influenced by the tumour-immune microenvironment, immune escape mechanisms, and limited ability to identify potential responders who should receive immunotherapy. There is a high unmet need for new innovative approaches to immunotherapies to battle not only CRC but cancers in general. This review will look at the work that is currently addressing this need through techniques such as genome-wide CRISPR/Cas9 screens, targeted in vivo models, and co-culture organoid systems. The goal of these approaches is to discover ways to reverse immunotherapy resistance to make this a more widely effective treatment for patients with CRC.

## 1. Introduction

Cancer is a leading cause of death worldwide. The most recent data collected by the World Health Organization (WHO) in 2022, estimated over 20 million new cases of cancer were diagnosed and 9.7 million deaths occurred worldwide. Colorectal cancer (CRC), which includes cancers of the colon (large intestine) and the rectum, accounts for approximately 10% of all cancer cases and is the second leading cause of cancer-related deaths worldwide [1]. Early-stage tumours have a high (91%) 5-year survival rate, but this drops dramatically to 13% in the metastatic setting, of which ~70% metastasize to the liver [2,3,4]. Of newly diagnosed patients, 35% already have metastatic disease, and a further 20–50% of patients will develop metastasis [5]. The dismal 5-year survival rate for late-stage disease is mainly due to the lack of effective therapies [6].

Colorectal cancer treatment is primarily tailored to the tumour stage. For localised colon cancer (stage I–III), surgical resection with adequate margins and regional lymphadenectomy is potentially curative, with adjuvant chemotherapy recommended for all stage III and selected high-risk stage II patients. Management of metastatic disease (stage IV) relies primarily on systemic chemotherapy, paired with biologics such as the angiogenesis inhibitor bevacizumab or, for RAS/BRAF wild-type tumours, the EGFR inhibitors cetuximab or panitumumab. Oligometastatic disease may be treated with surgical metastasectomy or ablative techniques to achieve long-term survival. The PD-1 immune checkpoint inhibitor pembrolizumab is indicated as a first-line monotherapy for unresectable or metastatic tumours with microsatellite instability (MSI) or mismatch repair deficiency (dMMR). Nivolumab (PD-1 inhibitor) is approved as a single-agent therapy in MSI/dMMR metastatic CRC following progression on fluoropyrimidine, oxaliplatin, and irinotecan. Combination nivolumab and ipilimumab is approved for unresectable or metastatic MSI/dMMR colorectal cancer, marking the first dual PD-1/CTLA-4 immune checkpoint blockade (ICB) regimen in this setting [7].

## 2. Colorectal Cancer: Immunotherapy’s Black Sheep

While many solid tumours and blood-based cancers have shown an enduring response to immunotherapies, this promise has not been realised for colorectal cancer. Whilst MSI/dMMR is predictive of the response to immunotherapy, this is observed in only 5% of patients in the metastatic setting. These tumours have defective DNA mismatch repair, which in turns leads to a high tumour mutational burden (TMB). This leads to the accumulation of neoantigens, which are recognised by the immune system in an attempt to control the cancer. Tumours with a mutation in the polymerase gene POLE are microsatellite stable but demonstrate a very high TMB and may also respond to immunotherapy [8,9]. A high density of CD3^+^/CD8^+^ tumour-infiltrating lymphocytes (termed Immunoscore) and positive staining for PD-L1 expression indicate pre-existing anti-tumour immune activity and can be used as biomarkers to predict response. While these are not widely used in CRC patients, due to the limited use of immunotherapies, circulating tumour DNA (ctDNA) and exosomes are increasingly being recognised as valuable liquid biopsy tools for monitoring the response to immunotherapy in cancer patients [10,11,12,13]. ctDNA, released into the bloodstream from tumour cells, provides a dynamic snapshot of the tumour burden and mutational landscape, with declining levels often correlating with treatment success and rising levels signalling early resistance or progression. Exosomes, which are small extracellular vesicles secreted by the tumour and immune cells, carry proteins, RNA, and DNA that reflect both tumour biology and immune activity [14,15]. Together, ctDNA and exosomes offer minimally invasive biomarkers for predicting and detecting immunotherapy outcomes. Since the majority of colorectal cancers are negative for these biomarkers, efforts are now focused on identifying new biomarkers and both intrinsic and extrinsic factors that may provide therapeutic targets for reversing immunotherapy resistance.

In colorectal cancer, particularly microsatellite-stable (MSS) tumours that often fail to respond to conventional checkpoint inhibitors, T-cell immunoglobulin and mucin-domain containing-3 (TIM-3), lymphocyte activation gene-3 (LAG-3), and T-cell immunoreceptor with Ig and ITIM domains (TIGIT) have gained attention as novel inhibitory receptors. These molecules have been shown to contribute to T-cell exhaustion and immune evasion, dampening anti-tumour immunity within the tumour microenvironment [16,17]. Studies suggest that blocking TIM-3 can restore T-cell function, while LAG-3 inhibition synergizes with the PD-1 blockade to enhance anti-tumour responses [18]. TIGIT, which interacts with CD155 (also known as the poliovirus receptor (PVR)), suppresses T-cell and NK cell activity, and antibodies such as tiragolumab are being tested to counteract this pathway [19]. Collectively, these checkpoints represent the next-generation immunotherapy targets that may expand treatment options for colorectal cancer patients who do not benefit from current immunotherapies.

### 2.1. Cancer Cell Intrinsic Factors That Influence Immunotherapy Response

**Antigen presentation:** The killing effect caused by ICB relies on the targeting capacity of T cells to recognise the tumour via its major histocompatibility complex (MHC) surface proteins. MHC Class I proteins are recognised by cytotoxic T cells and are essential for mounting an effective T cell response. Mutations have been observed in genes encoding proteins in the MHC including MHC molecules and Beta-2-microglobulin (β2M), which may confer ICB resistance [20,21]. The staphylococcal nuclease and Tudor domain containing 1 (SND1) is upregulated in colorectal cancer which leads to the ER mediated degradation of MHC proteins, which could also mediate ICB resistance [22,23].

**Signalling aberration:** Colorectal cancer arises when multiple regulatory circuits that normally keep cell growth in check become derailed, granting tumours the core abilities needed for development and progression. The Wnt/β-Catenin pathway is one of the most frequently altered pathways in colorectal cancer, with direct effects on both CD4 and CD8 T cells. Mutations in this pathway have been shown to affect CD4 T cell function through IL-17 and IFNγ signalling, promoting tumour growth [24]. The chemokine CCL4 is highly expressed in colorectal cancer and has been seen to affect CD8 T cell cytotoxic function, potentially through the Wnt/β-catenin pathway [25,26,27]. The MAPK/ERK cascade is activated in half of all CRC and acts as a central conduit for growth and differentiation signals. Mutations in critical nodes not only fuel aberrant MAPK signalling but also serve as valuable biomarkers for personalised therapy. The PI3K/AKT/mTOR pathway is also very important for cell growth, survival, and metabolic activity when it is hyperactivated. Cancer cells hijack this cascade to secure a continuous supply of energy and biomass, tipping the balance heavily in favour of unchecked proliferation. Signalling through transforming growth factor-β (TGF-β) has a paradoxical role in CRC. Early on, it acts as a tumour suppressor and restrains neoplastic transformation by slowing cell division. As the tumour evolves, however, within the tumour microenvironment, TGF-β fosters immunosuppression, by inhibiting cytotoxic T cell and NK cell activity, while simultaneously enhancing regulatory T cell and myeloid-derived suppressor cell function. At the cellular level, TGF-β signalling induces an epithelial-to-mesenchymal transition (EMT), a process in which epithelial cells lose polarity and adhesion, acquire mesenchymal traits, and gain migratory and invasive capabilities. This EMT not only facilitates metastatic spread but also contributes to therapy resistance by creating a more aggressive, stem-like phenotype. By reshaping both immune and stromal compartments, TGF-β acts as a central driver of tumour progression, making it a possible target for therapeutic strategies aimed at overcoming metastasis and improving immunotherapy efficacy [28,29,30].

### 2.2. Cancer Cell Extrinsic Factors That Influence Immunotherapy Response

The tumour microenvironment (TME) comprises stromal and immune cells that effect cytotoxic T cell infiltration and activity and is also influenced by the microbiome.

**Stromal cells:** Unlike other stromal cells such as immunosuppressive cells and vascular endothelial cells, cancer-associated fibroblasts (CAFs) play unique roles in the development of physical barriers for drug penetration through extracellular matrix (ECM) production, suppression of anti-tumour immunity, angiogenesis, and enhancement of cancer cell migration. These functions of CAFs contribute to therapeutic resistance and poor treatment outcomes [31,32]. ECM is the non-cellular constituent found in all organs and tissues. The major ECM proteins include collagen, proteoglycan, and elastin. CAFs contribute to the dynamic remodelling of the ECM by secreting matrix metalloproteinases (MMPs) [33,34]. The ECM not only mechanically supports cancer tissues but also acts as a reservoir for growth factors, releasing them to cancer cells under specific conditions. It also acts as a scaffold that promotes cancer cell interactions with growth factors. Furthermore, ECM re-modelling (via degradation) modulates the interaction of a scaffold with cancer cells and regulates the cellular metabolism and signalling pathways [35,36].

**Immune milieu:** Regulatory T cells (Tregs), T Helper 17 cells (Th17), myeloid-derived suppressor cells (MDSC), and tumour-associated macrophages (TAM) can contribute to an immunosuppressive tumour microenvironment. CD4 Helper T cells include Tregs and Th17 cells that regulate immune activity. Tregs express immune checkpoints on their surface such as PD-L1 and CTLA-4 and secrete molecules such as TGF-β, IL-25, and IL-10, which together can suppress cytotoxic T cell function [37]. They can also express CD25 and convert ATP into Adenosine, restricting the availability of nutrients for T cells. CD25 acts as a gatekeeper for IL-2 signalling, directly shaping ATP metabolism in T cells. Effector T cells depend on CD25 to fuel glycolysis and expansion, while Tregs use constitutive CD25 expression to monopolise IL-2, sustaining oxidative phosphorylation-driven suppressive functions. This balance is central to immune regulation and a key target in immunotherapy [38,39]. Th17 cells release chemokines that influence cytotoxic T cell and neutrophil infiltration, and pro-tumourigenic molecules including IL-6 and VEGF [40]. MDSCs are an undifferentiated population of immune cells with potent immunosuppressive activity that originate from the bone marrow’s myeloid progenitors. They are involved in many pathological conditions, such as cancer, and work by suppressing the immune system’s response, particularly T cells, and by promoting processes like tumour growth, angiogenesis, and metastasis. MDSCs are being studied as potential therapeutic targets in cancer treatment [41]. These cells use L-arginine through Arginase-1 (ARG1) inducible nitric oxide synthase (iNOS). These processes deplete the TME of L-arginine, a key metabolite for T cell function. Specifically, nitric oxide made from iNOS directly hinders T cell activity [42]. Macrophages may reversibly polarise to M1 or M2 states, with M2 being the more immunosuppressive state. These are the predominant TAM populations and can reduce the function of cytotoxic T cells. Similarly to MDSC, they express ARG1 and iNOS, but also express PD-L1 and secrete IL-10 that directly inhibit the activity of T cells. TAMs are also capable of secreting angiogenic factors, including VEGF, MCP-1, MIP-1a, and MIP-2a, which promote infiltration of other immunosuppressive cells [43,44].

**Gut microbiome:** The gut microbiota plays a critical role in maintaining homeostasis in the colon. The microbiome has been shown to promote colorectal tumorigenesis by increasing Th17 activity by the production of bacterial IL-17 [45]. Disrupted microbial diversity or an abundance of individual species may promote tumorigenesis and influence the immunotherapy response. For example, *Fusobacterium nucleatum* is more abundant in ICB non-responders. Furthermore, a faecal microbiota transfer (FMT) from responding patients into non-responders confers sensitivity to ICB. Conversely, performing an FMT from donors that were non-responsive does not confer sensitivity. The microbiome is so important it has been shown taking antibiotics prior to treatment can reduce the immunotherapy response significantly due to the dysregulation of the microbiota [46]. Recent studies identified 4-hydroxybenzeneacetic acid (4-HPA) as a key immunomodulatory metabolite secreted by gut microbiota leading to enhanced colorectal cancer progression [47]. The gut microbiota in colorectal cancer is significantly altered, with reduced biodiversity and abundance of certain species. CXCL3 release stimulated by 4-HPA directly recruits polymorphonuclear-MDSCs which in turn attenuates CD8^+^ T cell infiltration and activity. In a mouse setting, gut microbiota was found to synergize with anti-PD1 therapy, where combining the treatment with broad spectrum antibiotics diminished tumour control [48]. Certain bacterial species, such as that of *Akkermansia muciniphila* and *Faecalibacterium prausnitzii*, were found to induce transcriptomic differences in immune cells. These changes translated into significantly altered general immune cell groups as well as specific T cell and myeloid subsets. The effects of gut microbiota in immune mediated control of tumours are complex, and it is clear certain species can be beneficial while others detrimental.

## 3. CRISPR/Cas9 Screens

Even though clustered regulatory interspaced short palindromic repeats (CRISPR) were discovered in prokaryotes in 1987 [49], it was another 26 years before the CRISPR/CRISPR associated protein 9 (Cas9) system was developed into an invaluable tool for genome editing in eukaryotes [50]. CRISPR screens use the CRISPR/Cas9 system to disrupt, activate, or repress genes across the genome to identify their functions. The robust platform of high-throughput CRISPR/Cas9-mediated screening, whether implemented in cancer cells, immune cells, or Cas9-expressing mice, can facilitate the identification of genes associated with antigen presentation, immune evasion, immune checkpoint regulation, T cell-mediated tolerance, and immune cell activation. In the field of cancer immunotherapy, the application of CRISPR/Cas9-based genetic perturbation screening has facilitated the identification of genes, biomarkers, and signalling pathways that govern cancer immunoreactivity, as well as the development of effective immunotherapeutic targets, thereby fostering the development of immunomodulatory drugs and optimal therapeutic strategies [50,51,52] (Figure 1).

### 3.1. CRISPR Methodolgy

CRISPR/Cas9 screening is a high-throughput approach for the systematic disruption of genes across the genome to uncover those that drive or suppress a given phenotype [53,54]. By pairing a library of single guide RNAs (sgRNAs) with the Cas9 nuclease, targeted DNA breaks can be introduced that result in specific gene knockout. Pooled formats enable the simultaneous interrogation of thousands of genes in one experiment. Arrayed CRISPR screens use individual sgRNAs per well (or sample) rather than in a pool. This approach increases specificity but is more costly and labour intense. Library design and construction begin by defining the screen’s scope, either a genome-wide library covering roughly 19,000 protein-coding genes with multiple sgRNAs per gene (usually between 3 and 10) or a focused set to test a specific hypothesis. sgRNA sequences are chosen to target early exons and scored by predictive algorithms for high on-target activity and minimal off-target similarity. Once selected, guides are cloned into lentiviral vectors which can co-express Cas9. Alternatively, Cas9 can be added using a second vector, using mRNA or as a protein as well as a unique barcode, enabling pooled delivery and downstream tracking. Deep sequencing-based quality controls confirm uniform sgRNA representation and correct insertions before the library is stored as plasmid pools, glycerol stocks, or viral particles. Advances like high-fidelity Cas9 variants, dual-sgRNA constructs, and CRISPR interference (CRISPRi) or CRISPR activation (CRISPRa) libraries further boost knockout efficiency and expand the range of functional screens.

CRISPR/Cas9 screens can be categorised by both the perturbation modality and the selection strategy: classic knockout (CRISPRko) screens introduce frameshift mutations to disrupt gene function, while CRISPRi and CRISPRa use catalytically dead Cas9 fused to a repressor or activator domains to downregulate or upregulate target genes without cutting DNA. Selection-based screens are divided into negative (dropout) screens, where essential genes are depleted over time, and positive (enrichment) screens, which identify mutations that confer drug resistance or growth advantages. Focused libraries can be employed to target specific pathways, combinatorial or synthetic-lethal screens that knock out gene pairs, and tiling approaches that densely cover regulatory regions. Advances in single-cell readouts further enable linking each genetic perturbation to transcriptomic changes.

### 3.2. Three-Dimensional CRISPR Screens

Three-dimensional CRISPR screens have been shown to more faithfully reproduce in vivo tumour behaviour than traditional two-dimensional assays [55]. Expanding this approach to 3D intestinal organoids has further improved the modelling of human colorectal cancer. For example, Michels et al. carried out a tumour suppressor focused CRISPR screen in pre-malignant APC^−^^/−^; KRAS^G12D^ organoids both in vitro and in vivo, identifying TGFβR2 as the top suppressor along with additional known and novel regulators of tumour growth [56]. Genome-scale screening in organoids has been limited by technical challenges, including the large cell numbers required. To overcome this, Ringel et al. developed a method to capture sgRNA integrations from single organoids, dramatically reducing the input cells needed for comprehensive screens. Together, these advances pave the way for applying CRISPR screening across diverse organoid systems, opening new routes to dissect the genetics of human disease [57].

### 3.3. In Vitro CRISPR Screens

In vitro co-culture systems including primary tumour cells or organoids, cultured with immune cells, stromal cells, or even microbes, provide an opportunity to interrogate specific interactions between components of the tumour microenvironment. In these systems, either the primary cell or the other cell type can be screened using an appropriate CRISPR library. This approach can give insights into tumour–immune cell interactions and functions of specific cell types in a systematic way. Both loss-of-function and gain-of-function screens have been successfully applied in somatic cells using CRISPRi and CRISPRa in combination.

### 3.4. In Vivo CRISPR Screens

Integrating CRISPR technologies with in vivo cancer immunology studies offers a more biologically relevant model of tumour biology by enabling the observation of interactions between tumour and immune cells within their native tissue microenvironment. Recent research has showcased the effectiveness of in vivo CRISPR-Cas9 functional screens, particularly in models of acute myeloid leukaemia (AML) and chemoresistance. These studies have identified key regulatory genes, such as BCL2, BRIP1, and COPS2 that influence therapeutic responses [58,59]. Two primary strategies are employed for in vivo CRISPR screening. Indirect in vivo screening involves transplanting a pool of mutagenized cells into a recipient animal, typically subcutaneously, and direct in vivo screening delivers CRISPR components directly to the target organ using viral vectors (AAVs, lentiviruses, tissue-tropic viruses) or non-viral methods (nanoparticles, microinjection, electroporation). Although in vivo screening offers a more physiologically relevant context by targeting specific tissues, it is hampered by challenges such as low delivery and infection efficiency. As a result, the scalability of sgRNA libraries in direct in vivo applications remains limited.

### 3.5. Liminations and Mitigation Strategies

Despite revolutionising functional genomics, CRISPR/Cas9 screens have several limitations. These include off-target cleavage which can introduce spurious mutations, variable editing efficiency, and the frequent occurrence of in-frame indels, which may induce only partial knock out and reduce screen sensitivity. Essential genes are also hard to assess because a complete loss of these often causes cell death and dropout biases. Pooled screens in homogeneous cell lines overlook complex in vivo contexts like the microenvironment or immune interactions, and uneven sgRNA representation and library design gaps can generate false negatives. Standard loss-of-function approaches struggle to probe noncoding elements, gene dosage effects, or epigenetic regulation, necessitating complementary assays for a more complete functional map.

To overcome these common CRISPR/Cas9 screening pitfalls, the design and validation of high-quality sgRNA libraries should employ predictive tools and multiple guides per gene, alongside appropriate controls, to minimise off-target effects and dropout biases. Currently there are 375 pooled libraries that are available through Addgene as well as many commercial libraries. Pairing this with high-fidelity Cas9 variants or transient RNP delivery and optimised infection conditions ensures single-guide integration and uniform library coverage. Other approaches to overcome these limitations include by incorporating CRISPRi/a, and by testing diverse cell types or organoids through co-culture models. Robust analysis pipelines (like MAGeCK, MAGeCK-VISPR, ScreenBEAM or CERES) can correct for guide efficiency and copy-number effects. Comprehensive validation of candidate genes and pathways should be performed through individual functional assays, orthogonal methods, rescue experiments, and in vivo models. The integration of transcriptomic and/or proteomic data can highlight genes that can be directly therapeutically targeted or in combination with an existing treatment.

Potential antigens including Cas9, antibiotic resistance genes, inducible expression system components, and fluorescent proteins should be removed prior to a CRISPR screening with immunotherapies to avoid unintended immune recognition. The SCAR lentiviral vector system was developed to overcome this problem, which allows engineering of stable Cas9 and sgRNA expression, efficient and scalable vector delivery, and immunogenic antigen removal using Cre recombinase [60]. Immune recognition of CRISPR vector associated antigens may not be problematic in all model systems and has previously been identified ad hoc. Strong anti-vector immune responses are of particular concern when stable lentiviral transduction is experimentally required in vivo but may be less of a concern in simpler model systems such as organoid-immune cell co-culture models.

Although no approved drugs have yet emerged directly from CRISPR/Cas9 screens, these experiments have identified high-value targets (see Table 1 for colorectal cancer-related findings) that are now being pursued in drug development pipelines. An example of this includes genome-wide CRISPR knockout screens in MSI colorectal cancers that pinpointed the WRN helicase as a synthetic-lethal vulnerability, prompting the preclinical development of WRN inhibitors [61,62,63].

## 4. CRC Mouse Model Systems for In Vivo CRISPR Screens and Validating Targets

This section will give an introduction to the types of CRC mouse models available and provide an example of how these models have been used to investigate mechanisms, tumour drivers, and treatment effectiveness.

Animal models have been crucial for understanding disease initiation, progression, and testing therapies in a complex model system. There are several mouse models for CRC, each with benefits and limitations. CRC mouse models may not always faithfully replicate the progression of disease from adenoma to adenocarcinoma and subsequent metastasis, as seen in patients. Therefore, investigations of these stages require the use of several mouse models to examine the effect of any tested therapy on all stages of disease progression [58]. Despite this, mouse models allow the study of complex interactions between the tumour and the microenvironment, and how therapies may impact tumour growth [80].

Mouse models are invaluable to testing new CRC therapies. The standard of care for CRC patients is chemotherapy alone or in combination with targeted therapies such as bevacizumab (anti-VEGF), regorafenib (anti-TK), or immunotherapy (targeting MSI CRC). The most common chemotherapy regimen combines 5-fluorouracil (5-FU), leucovorin, and oxaliplatin, known as FOLFOX, and is routinely tested in these models. Mice have been determined to be quite sensitive to leucovorin, which can be excluded in these studies with a similar chemotherapy in mice [81].

There are three main types of CRC mouse models. These include those established by carcinogen exposure, genetic manipulation, or the transplant of human or mouse cell lines, tissue, or organoids into recipient mice.

### 4.1. Carcinogen-Induced Models (CIMs)

CIMs involve the administration of a known carcinogen to mice to initiate tumours (recently reviewed [80,82]). These carcinogens, such as methylazoxymethanol azoxymethane (AOM), methylazoxymethanol acetate, etc., can be given either as a feed ad libitum, oral gavage, or by peritoneal injection, causing intestinal mutation to initiate tumour development [82,83]. These are also regularly combined with dextran sodium sulphate to induce inflammation creating a colitis model. These models have been instrumental in 80 years of CRC research and reflect the primary disease progression similar to patients [82]. Variance in the incidence of tumour development is caused by mouse gender, age, microbiota, and genetic background. CIMs generally cause the development of small intestinal tumours and not the colon as seen in patients [82]. Typically, CIMs take a long time before cancer development (8–30 weeks) [83] which makes choosing a therapeutic window difficult. Therefore, CIMs are better used for research investigating the effect that diet and dietary supplements, chemopreventive interactions, and the gut microbiome have on CRC.

The investigation of the effect of a high-fat diet (HFD) was conducted in colitis-associated cancer caused by a combination of AOM/DSS treatment in mice. HFD increased mouse weight loss and reduced survival caused by hyperplasia and multiplicity of tumours. The addition of an herbal supplement derived from *Aster glehni* was able to reverse inflammation and reduce cancer-related death, prevent weight loss, and reduce spleen enlargement in mice [84].

### 4.2. Genetically Engineered Murine Models (GEMMs)

GEMMs focus on creating mice with specific mutations or deletion of either tumour suppressor genes (such as *Apc*, *p53*, *Smad*), a gain of function oncogenes, (e.g., Kras, *Src*, *C-myc*), DNA repair genes (e.g., *MSh2*), or a combination of these mutations to drive the formation of tumours [80,82,83,85]. This model tumour formation can be restricted by a constitutively expressed or inducible Cre under the control of intestine-specific promoters (such as *Villin* or *Lgr5*) to target flox sites to specifically delete genes in the epithelial cells of the large and/or small intestine. GEMM tumours can form either spontaneously with age or they may be accelerated by a carcinogen [80,82]. Depending on the mutation and induction method, extensive work has been completed to identify models of different types of CRC (e.g., adenocarcinoma), location (large versus small intestine), and reflecting different stages of cancer development and the ability of the primary tumour to metastasize [83,85].

The foundation GEMM for CRC is the APC^Min/+^ mouse model. Although this mouse could also be described as a CIM as it was initially caused by the administration of N-ethyl-N-nitrosourea leading to a nonsense mutation in codon 380 of the *Apc* gene. This strain and more targeted mutations into different strains are all homozygous embryo lethal, but heterozygous mice were used to establish the first model for multiple intestinal neoplasia (Min) [82]. After developing Min, mice develop familial adenomatous polyposis and eventual colorectal cancer (recently reviewed [80,86]). This model has been critical in understanding the initial tumour drivers of CRC. APC has been identified as a CRC driver, and loss/mutation causes upregulation of the Wnt pathway and accumulation of B-catenin leading to abnormal cellular signalling [72]. The APC^Min/+^ mouse model was instrumental in research in mismatch repair (MMR), identifying other tumour-promoting genes, and drug resistance by crossing the mice with other GEMMs [80,86].

### 4.3. Transplant Models

Other types of CRC models rely on the transplant of syngeneic or xenograft (cross-species) tissue, tissue-derived organoids, or cell lines into mice to establish CRC. These models can establish a tumour in the desired location based on needs either by transplanting orthotopically or within a location that reflects primary or metastatic CRC. Injecting samples orthotopically into the caecum, distal colon, or rectal mucosa provides a relevant microenvironment for tumorigenesis to develop but can be technically challenging and requires a high level of skill. Injecting subcutaneously into the mouse flank is an alternative low-technical method to establishing primary mimicking CRC but may not be as clinically relevant and is unable to seed metastases [82,87]. Orthotopic transplant may metastasise into neighbouring or distal sites dependent on the characteristics of the transplanted material. As such, injection into the metastatic location of CRC may overcome limitations of the former and be capable of modelling late-stage metastatic disease. Techniques used to model metastatic growth in mouse models for CRC include injecting directly into the liver or into the portal vein for direct drainage into the liver or intra-splenic injection for dissemination to the liver. Similarly to an orthotopic transplant, these are surgical techniques requiring a high skill set. Metastasis of the peritoneal cavity can also be achieved by intraperitoneal injection, and lung metastasis can be achieved via tail-vein injection [82,85].

Human transplants for these assays could be cell line-derived, patient-derived, or patient-derived organoid xenografts. There are over 100 different curated human CRC cell lines with known mutations that can be transplanted into immunocompromised mice. Recipient mice for the xenograft are numerous, and all have different mutations in the immune system with different levels of immunodeficiency, but most lack T, B, and NK cells and may have an altered myeloid function to avoid rejection of the human cell transplant [82,88]. The more immunocompromised the mouse, the higher the take rate of the tumour [88]. Consideration of which immunocompromised mouse line to use could affect the mouse modelling. For instance, the C:B:-17 *scid* and strains derived from this mouse strain carry a homozygous mutation of the *Prkdc* gene, which is responsible for DNA repair, making the mouse extremely susceptible to radiation and DNA-damaging reagents [82].

Common human CRC cell lines used in mouse modelling include MSI lines HCT-116 and HT-15 and MSS lines SW620 and HT-29 [83,89]. As there is a vast amount of information on these cell lines, a great amount of information can be obtained using these models. These lines can be transplanted into mice with and without Matrigel to improve take rates in models to establish primary or metastatic disease. There are concerns that the cell lines do not reflect patient data due to the long time they are in tissue culture, causing genetic drift over time [90]. Similarly, as xenograft CRC models are in immunocompromised mice, immune studies especially based on T cell function are not possible [85]. One team used the varying immunocompromised nature of the nude mouse compared to NSG (NOD scid gamma) to investigate the formation of neutrophil extracellular traps leading to tumour cell apoptosis. This team found that the combination of 5-FU and telaglenastat (CB839), a therapy for glutamine-dependent CRC, caused the release of neutrophil extracellular traps in nude mice but not NSG transplanted with a human cell line xenograft which caused apoptosis of tumours through the release of Cathepsin G [91].

Mouse cell lines commonly used for syngeneic immunocompetent CRC models include the MC38 (C57bl/6) and the CT26 (Balb/c) and are transplanted into mice of the strain of cellular origin [92]. MC38 reflect MSI-H tumours containing a mutation in the MSH3 genes [93], whilst CT26 reflect MSS tumours containing a Kras^G12D^ mutation [94]. The CT26 are the more versatile model able to metastases from intra-caecal injection to cause metastasis [95,96]. These cell lines can also be genetically mutated to knock out or over express genes and can have reporter genes added as required [82,95,96]. Although concerns have arisen relating to these syngeneic cell lines being quite sensitive to anti-PD-1 or anti-CTLA-4 unlike most CRC patients [92,97]. An alternative line, Colon 26 (MSS, *Kras* mutant, Balb/c), is less sensitive to anti-PD-1 therapy but still sensitive to CTLA-4 inhibition. This sensitivity variance was caused by a lower infiltration or exclusion of immune cells into the tumour coupled with a high level of Wnt signalling making the Colon 26 tumour resistant to anti-PD-1 therapy [92].

Subcutaneously implanted MC38 and CT26 have been treated with 5-FU and oxaliplatin which increased the activity of tumour infiltrating CD8 T cells and prevented exhaustion of the cytotoxic T cells, as they demonstrated lowered expression of inhibitory receptors of PD-1 and TIM-3 [81]. A subsequent combination of FOLFOX with anti-PD-1 therapy promoted tumour regression that is unattainable with monotherapies [84]. FOLFOX combined with anti-PD-1 was found to increase tumour infiltration of active cytotoxic CD8 T cells by overcoming adaptive immune resistance by the tumour [98].

A xenograft of patient tumour tissue into mice is a useful tool to study the effectiveness of anti-cancer agents and the biology of these resultant tumours. Total tumour tissues form a heterogeneous cell population which can reflect the patient’s tumour biology, but issues arise with the low take rate of the xenograft in mice and increased costs associated with these experiment types. However, benefits include that these human-derived tumours can then be maintained in the mice to seed future experiments [99]. Tumours have also been established using malignant ascites or circulating tumour cells [83]. The generation of new CRC lines is possible using this method but is reported to only be possible with 10% of samples [100].

Patient tissue implanted into mice can expand to then seed multiple mice. However, this does have a lower take rate based on the variability of samples, time to implantation, the use of Matrigel, and the original growth of original tissue [100]. Mice were treated with the anti-EGFR antibody cetuximab and responded with similar rates as in the clinic. This identified a HER2 amplification in patient tissue wild-type cells for EGF pathway mutation markers but were known to be resistant to EGFR therapy. A combination of cetuximab and anti-Her2 therapy in these resistant xenografts was able to increase tumour regression in mice [99].

More success and higher take rates have occurred by creating human patient-derived organoid (PDO) cultures for subsequent transplantation into mice [83,101]. These cultures are created from isolated CRC progenitor cells grown in 3D multicellular aggregates in a media that is rich with cytokines and growth factors to support PDO proliferation [102]. Increased take rates may be due to the filtering of the sample through selective pressure to exclude any organoids not fit for continual growth. PDOs mimic drug sensitivity to patient cancer but maintain a heterogeneous population of cells reflecting the genetic, transcriptomic, and histological traits of the patient [83,101,102,103]. PDOs can be transplanted into mice to reflect primary (subcutaneously), or metastatic models (intra-spleen, intra-portal vein). When CRC organoids were implanted intra-splenically, it was found that only those from a metastatic origin could disseminate and establish liver metastases while organoids derived from primary CRC samples remained and developed tumours in the spleen. These PDOX mice were treated with a 5-FU and oxaliplatin combination and the response reflected what was seen in the clinic for these patients [101].

Mouse organoid models have been established using CRISPR genetically modified normal colon cells to mutate CRC driver genes such as *Apc*, *Tp53*, or/and *Kras* [83,104,105]. Additionally, organoids derived from a tumour carrying these mutations can be cultured [104]. These models are flexible to test mechanisms based on gene manipulation and test therapies. Organoids were isolated from mice containing a variety of different combinations of gene mutations Apc^LoxP^, LSL-Kras^G12D^, Tgfβr2^LoxP^, and p53^LoxP^ under an inducible Lgr5 promoter to determine if metastatic tumours could be generated by gene mutations. It was found the more genes mutated, the more likely the organoids could metastasise to the liver and lungs. Organoid transplanted mice were also identified to be extremely sensitive to Galunisertib (TGFβR1 inhibitor) therapy with a reduced tumour size if implanted into the caecum or a reduced number of liver metastases. LAKTP organoids were also resistant to anti-PD-1 therapy unless combined with Galunisertib therapy [104].

Recent work has focused on humanising the immune system of immunocompromised mice for subsequent CRC studies targeting immune reactivation such as checkpoint blockade. This has been improved with recent advances in human gene knock-in models (such as the NSG-SGM3-IL15) better able to support all human immune cells in mice [82,88,90]. The human immune system is established by the transplantation of hematopoietic stem cells derived from bone marrow, G-CSF mobilised blood or cord blood CD34^+^, or patient-derived or donor PBMC [82]. Bone marrow and cord blood CD34 progenitor cells can be purchased commercially with and without HLA-typing to match the tumour samples, making this a feasible option. Mice are transplanted with the immune cells, which allows the development of human immune cells to then receive the cancer implant (cell line, PDX, PDO) followed by therapy application. This model may confirm that mouse therapy will work on patient samples with a human immune system under controlled experimental parameters and is especially important for testing potential immunotherapies.

In a study where the NSG mice immune system was humanised using patient PBMC, anti-PD-1 therapy against matched colorectal liver metastatic samples implanted subcutaneously demonstrated similar responses to the clinic [106]. The use of PBMC from an allograph source was not able to reflect patient responders. Using patient PBMC did eventually cause graft-versus-host effects in the mice but did allow a sufficient experimental window of up to 50 days if a smaller number of PBMC were implanted in which to test the immunotherapy [106].

Another team utilised the NRG-HLA-A2 mice that contain the human HLA gene to improve T cell education in human transplanted immune cells. These mice were then transplanted with cord CD34^+^ cells before receiving the HCT-116 cell transplant subcutaneously. Mice were then treated with a combination immunotherapy of human anti-CTLA-4 and anti-PD-1 with the addition of an oncolytic bovine herpesvirus plus mitomycin, which is a method to reactivate cold tumours and improve immunotherapies [107]. Dual immunotherapy was relatively ineffective, but the combination of immunotherapy with the oncolytic bovine herpesvirus was sufficient to stall tumour growth. This effect was due to a reduction in Treg cells with an increase in cytotoxic T cells and a reduction in exhaustion marker Tim3 [107].

There is a broad range of mouse models of CRC. Each has benefits and limitations that will need to be carefully considered, and potentially more than one model may be needed for validation.

## 5. In Vitro Co-Culture Systems

Patient-derived cancer organoids form 3D heterogenous epithelial complexes that represent multiple epithelial cell types of the colon crypt and villi. The development of these organoids has been imperative in advancing the potential application of personalised therapy [108,109]. However, organoid culture on their own lack the additional cellular components that comprise the tumour microenvironment (TME). The TME forms a dynamic platform by which the various cell types, including immune, stromal, and vascular cells, and non-cellular components including extracellular matrix, surround and support the cancer cell’s growth. Signalling and interactions between these components dictate the behaviour of cancer cells in directly influencing their initiation success, metastatic potential, and response to drug treatment [110,111].

More recently, in vitro studies have described sophisticated methodologies that include the culturing of certain cellular components of the TME with human-derived CRC organoids to form co-cultures [112,113]. These co-cultures aim to further recapitulate in vivo biology and therefore provide a more physiologically comprehensive platform from which to investigate cancer cell–TME dynamics to potentially identify relevant therapeutic targets and effective therapies for patients with CRC. Varying methodologies for co-culture have been described. Organoids may be seeded in basement membrane extract (BME) domes with TME cells either included [114] or dispersed in the overlaid media [115], or both organoids and TME cells can be seeded in suspension with low percentage BME [116]. Alternatively, transwell inserts may be used where the organoids and TME cells are prevented from cell–cell contact by a porous membrane which allows cytokine release and signal transduction to be measured. Organoids seeded in a monolayer and coated with BME on the porous membrane allows access to both the apical and basolateral surface for comprehensive analysis [115].

Organoid technologies provide an efficient platform for which to assess drug screening applications. As co-culture with TME components potentially provides a more accurate representation of the tumour biology, this extended application could provide more detailed investigations of mechanisms of drug reactivity and/or resistance and may uncover effective therapeutic approaches.

Immunotherapies have provided enduring responses for many cancer types; however, the majority of CRC remain refractory to this treatment as they typically downregulate their immunosensitivity. By co-culturing immune cells with CRC organoids, the extent and mechanisms of this immune dampening and subsequent lack of response to immunotherapies at a personalised level may be elucidated. This would be highly informative for predicting which patients may have retained sufficient levels of immunoactivation to potentially benefit from receiving immunotherapy, causes of potentially reduced immunosensitivity, and strategies that aim to increase the response to immunotherapy treatment at a personalised level [110,113].

### 5.1. Co-Culturing CRC Organoids with Lymphocytes Derived from Peripheral Blood Mononuclear Cells (PBMCs)

Dijkstra et al. described the isolation of autologous lymphocytes from PBMCs of CRC patients to determine immunoactivity levels and assess the extent of immune mediated cancer cell killing at a personalised level [113,116]. Cancer organoids were pre-treated with IFNγ to promote antigen presentation, and lymphocytes were pre-activated following two weeks of incubation with autologous organoids and co-stimulatory CD28 for antigen recognition. This platform allows the expansion of tumour reactive lymphocytes which could be utilised for developing T cell-based therapies on an individual basis. In addition, this co-culture system has been used to confirm the extent of T cell reactivity in CRC patients following neoadjuvant immunotherapy which indicates this methodology may also be predictive in identifying patients who show effective responses to receiving immune checkpoint blockade therapies [117]. Wang et al. [118] co-cultured microsatellite stable CRC organoids with autologous CD8^+^ cells derived from PBMCs and exposed the co-cultures to bacteria *Fusobacterium nucleatum* (Fn) which is a known CRC-related pathogen and produces butyric acid as a metabolite or butyric acid alone. The co-cultures were then treated with anti-PD1 immunotherapy and the cancer organoids within the co-culture that were exposed to Fn, or butyric acid, underwent significant apoptosis compared to those without Fn or butyric acid. This was due to the butyric acid relieving the exhausted state of CD8^+^ immune cells which then made them more immunoactive and responsive to checkpoint immunotherapy. Overall, this study highlights the presence of Fn as a potential biomarker for enhanced reactivity to immune checkpoint blockade. Further investigations into the efficacy of combining butyric acid with immunotherapy to improve immune cell mediated cancer cell killing may be warranted.

### 5.2. Co-Culturing CRC Organoids with Tumour Infiltrating Lymphocytes (TILs)

Co-culture utilising isolated TILs, which are present within the cancer microenvironment, may be more representative of the tumour biology than lymphocytes derived from PBMCs. TILS can develop specific immunological reactivity against cancer cells due to their exposure to tumour-associated antigens and cytokines within the TME [119]. A study investigating neoadjuvant chemoradiation treatment in rectal cancer patients found a correlation with the extent of cancer organoid killing when in co-culture with autologous TILs and response to treatment, indicating that results of co-culture with TILs could be informative in predicting the response to such treatment types and mitigate unnecessary treatment side effects for patients whose co-cultures did not show an effect [120]. Furthermore, a study that treated melanoma organoids with Navitoclax (anti-apoptosis inhibitor) before placing the organoids in co-culture with TILs, found the cytotoxicity levels of TILs subsequently increased. Further studies that investigate whether this is synergistic with combined immune checkpoint blockade may be warranted [121].

CRC organoids provide a source of cancer-specific antigens, and co-culture with autologous TILs has also provided a platform by which these tumour-specific TILs can be enriched and expanded upon, similarly to that seen with lymphocytes derived from PBMCs [113,116]. However, the numbers of tumour-specific T cells from this TIL co-culture method were twice as high as that from the PBMC co-culture. This rate of expansion allowed for the subsequent isolation of tumour-specific T cell receptors and potential development of personalised TCR-T cell therapy [122]. A further study compared the cancer cell killing effects of T cells derived from PBMCs or from TILs. Both were co-cultured with CRC organoids from patients with MSI CRC who had local inflammatory conditions and were receiving anti-PD-1 therapy. The study identified greater cancer cell killing with PBMC than TIL-derived co-cultures, indicating an inhibition of the local immune response. It was concluded that the presence of local inflammatory states can provide resistance to immune checkpoint inhibition which has direct implications in identifying clinical features that confer poor responses to immunotherapy [123].

### 5.3. Co-Culture with Other Immune Cells

Natural killer (NK) cells are an important component of immune cytotoxicity and can synergistically interact with cytotoxic T cells to mediate cancer cell killing [124]. NK cells have been co-cultured with breast cancer organoids in a metastatic setting to reveal NK-directed druggable targets that upon treatment reduced metastatic potential [125]. A study of CRC-derived spheroids co-cultured with NK and T cells revealed the potential efficacy of a bispecific antibody against an NK cell activating receptor which resulted in increased NK cell activation and anti-tumour potential [126].

To fully understand CRC–TME interactions, CRC organoids were co-cultured with tumour-associated macrophages (TAMs) which resulted in a pro-tumourigenic and immunosuppressive gene expression programme. This reflected what was observed in vivo and confirmed this co-culture tool as a useful model to understand complex dynamics of the cancer niche [127]. Fang et al. co-cultured macrophages with T cells and CRC organoids and identified that SIRT1 overexpression led to increased macrophage differentiation to an immunosuppressive pro-tumourigenic phenotype [128]. Based on these results, future studies investigating the effects of SIRT1 inhibition within this platform may be warranted. Dendritic cells (DCs), which are antigen-presenting cells and essential for priming anti-tumour immune responses, have been co-cultured with metastatic CRC organoids in recent studies. Results demonstrated how mCRC organoids can disrupt DC function sufficiently for them to exert pro-tumourigenic effects on the TME [129]. Subsequent investigations utilising the same co-culture platform identified that the inflammatory mediators, PGE2 and IL-6, partially contributed to tumour-inducing DC plasticity. Additionally, antagonists against these reduced the pro-tumour DC population and may ultimately have implications for future clinical application [130]. Co-culturing techniques have included harvesting conditioned media from myeloid-derived suppressor cells (MDSCs) with T cells co-cultured with CRC organoids in an autologous setting. This helped to identify novel cancer-specific markers of immunotherapy resistance to further predict responders to immunotherapies [131].

### 5.4. CAR-T Cell Therapy Co-Culture

Chimeric antigen receptor-T (CAR-T) therapy can be a highly effective therapeutic whereby a patient’s T cells are genetically engineered to enable them to recognise tumour-specific antigens and destroy the patient’s cancer cells when reintroduced back into the circulation. This therapy has proved very effective for haematological cancers; however, due to the typical immunosuppressive effects of the TME, it has been more difficult to gain significant efficacy in solid tumours [132]. Co-cultures of CAR-T cells with glioblastoma organoids enabled the quantification of CAR-T proliferation, infiltration, and cytokine release associated with CAR-T activation, which offers an important model to assess potential CAR-T efficacy and patient sensitivity [133]. CRC organoids were co-cultured with a CAR-NK cell line to assess CAR-NK cell-mediated cytotoxicity, representing a key step in modelling CAR-T therapy effectiveness for CRC at a personalised level [134].

### 5.5. Co-Culture with Stromal and/or Vascular Cells

Cancer-associated fibroblasts (CAFs) are the main stromal cell type providing a supportive and interactive environment for cancer epithelial cell proliferation and dissemination, and they can secrete immunosuppressive cytokines such as TGFβ and recruit pro-tumourigenic TAMs to the TME [135]. A recent study investigated the sensitivity of CRC organoids to standard of care drugs and found that in co-culture with CAFs the organoids had a higher drug resistance than single cultures of organoids alone. Additionally, the expression of signalling pathways of co-cultures were more similar to the originating tissue than single organoid cultures, indicating a greater representation of drug response with the addition of CAFs to organoids [136]. Other studies have shown that upon co-culture with CAFs, CRC organoids had reduced proliferation and favoured an EMT phenotype [137]. This study also showed that the co-cultures induced CAF-led changes to the matrix composition and demonstrated the potential of highly effective crosstalk between CAFs and cancer cells [138]. A recent study found co-culturing CRC organoids with autologous CAFs revealed the efficacy of drug combinations targeting paracrine growth signals and cyclin-dependant kinases. This provided an efficient platform to test tumour–stromal interactions for further therapy development [139]. Overall, these studies highlight the potential benefits of co-culture with CAFs to gain a more accurate measure of drug response and novel drug discovery using in vitro co-culture modelling.

Co-culture of CRC organoids with endothelial cells, which form the inner lining of blood vessels and lymphatics, can be useful to determine the induction of vascular tube formation. This can be indicative of the cancer’s vascularization potential and may also indicate patient sensitivity to anti-angiogenic therapies [140]. Hedayat et al. identified a resistance marker to regorafenib treatment which is a third line use multi-kinase inhibitor [141]. Regorafenib can be effective in promoting anti-angiogenesis to limit cancer progression but is associated with significant toxicities, which highlights the need to identify biomarkers for effective treatment. The study included endothelial cells to model the anti-angiogenic effects of regorafenib, as well as CAFs due to their ability to confer treatment resistance, in co-culture with CRC organoids. This co-culture modelling and transcriptomic analysis revealed upregulation of micro-RNA, MIR652-3p, was correlated with improved endothelial cell viability and was associated with resistance to regorafenib. Overall, this could serve as a much-needed biomarker for clinical benefit to regorafenib treatment.

### 5.6. Further Complex Co-Culture Modelling

Multiple cell type co-cultures involving more than one TME component with cancer organoids have been modelled to further represent the tumour niche, understand drug response mechanisms, and potentially identify promising therapeutic strategies. A study that cultured both CAFs and TAMs with matched organoids demonstrated that these co-cultures shifted towards pro-tumourigenic differentiation of TAMs compared to co-cultures with CAFs and organoids only. Therefore, this triple co-culture method allowed for more accurate responses to standard of care treatments to inform personalised strategies [142]. Li et al. found CRC organoids but not CAFs co-cultured with TAMs could induce immunosuppressive polarisation of TAMs towards a pro-tumourigenic SPP1^+^ phenotype in dual co-cultures [127]. When all three cell types were combined, the TAM phenotype was further enhanced, which is representative of SPP1^+^ TAMs in vivo, and potentially provided a platform for testing anti-SPP1 therapies that may reverse the immunosuppressive effects of TAMs [127,142]. Co-culture of CAFs and CD3^+^ lymphocytes with prostate cancer organoids revealed that by inhibiting the transcriptional co-activator, YAP1, CAFs transitioned to a more anti-tumourigenic and immunosensitive phenotype. This subsequently enhanced the therapeutic effect of combined anti-PD-1 application [143] and has important implications for exploring this strategy in CRC where YAP1 is typically upregulated and associated with an aggressive phenotype [144].

### 5.7. Advanced Platforms

The emerging “organoid-on-a chip” platforms utilise 3D microfluidic devices to provide a highly representative model of intestinal biology. These systems involve culturing the organoids on BME-coated porous membranes which are positioned on bioengineered flow-through chambers with channels in situ seeded with TME components that can recapitulate blood and/or luminal flow [145]. The dynamic microfluidics technology can allow for nutrient transfer, therapeutic delivery, and waste removal, as well as recapitulate peristaltic forces for highly accurate modelling. This platform has often been used to model tissue-specific vasculature with the inclusion of endothelial cells with and without additional TME components to assemble perfused microvascular networks within the 3D chambers. This can also allow in vitro testing of anti-angiogenic and/or immunotherapy modalities [146,147,148]. Miller et al. provided a novel device that mimicked all stages of the metastatic cascade whereby endothelial (HUVEC) cells were seeded with CRC and matched normal organoids. These were placed on opposite sides of a membrane in an air-liquid interface chambered fluidic device, and metastatic capabilities of organoids in response to variables such as hypoxia and glucose levels could be measured, and cellular competition between normal and cancer organoids could be assessed [149]. Microfluidics have been utilised to co-culture microbiota with intestinal organoids placed in a monolayer or as intestine-on-a-chip for a comprehensive analysis of host–microbial interactions which are crucial lines of investigation due to the key role that certain microbiota have in promoting CRC development [150]. Neal et al. introduced a methodology whereby CRC fragments were cultured in an air-liquid interface (ALI) method that enabled the culture of tumour epithelia whilst preserving the native tumour microenvironment [151]. This technique maintained the original tissue architecture and included differentiated immune cells as well as CAFs. The resulting ALI epithelial models successfully mirrored the activation of TILs in response to immune checkpoint blockade. However, a gradual decline in immune cells and fibroblasts was observed over 1–2 months after culture initiation, which may restrict the applicability of this platform for large-scale drug sensitivity testing.

Overall, these models of CRC patient-derived organoids co-cultured with autologous and/or allogenic components of the TME are instrumental in providing an accurate representation of the tumour niche within an efficient in vitro platform. The benefits of this in predicting patients who may benefit from such ground-breaking treatments, such as immunotherapies, and the potential in utilising these platforms to identify novel therapeutics on a personalised basis and capacity for biobanking purposes, warrants their continual development. However, the scope of this co-culture modelling is extensive and complex which calls for efforts to promote standardisation of techniques including cell type isolation protocols, media and matrix components protocols, and drug application protocols [152]. This would promote a reproducible and reliable methodology and directly translate to uncovering more effective therapies for patient benefit.

## 6. Conclusions

The convergence of CRISPR/Cas9 screens to identify essential genes for immunotherapy targets, well designed in vivo models, and in vitro co-culture systems utilising 3D organoids has transformed CRC research into a multidimensional discipline. CRISPR/Cas9 screens illuminate the hidden genetic dependencies that drive tumour growth and therapy resistance, while in vivo platforms validate these findings in the full biological context. Co-culture systems bridge the gap between reductionist assays and complex organisms, modelling the dialogue between cancer cells, fibroblasts, immune populations, and even the microbiome. Together, these technologies form a continuous pipeline: from high-throughput target discovery through CRISPR/Cas9 systems to physiological validation in animal models to mechanistic dissection in engineered microenvironments.

Looking ahead, the translational potential of this integrated framework is immense. By coupling PDO screens with tailored co-cultures, researchers can forecast therapeutic responses and anticipate resistance pathways before clinical intervention. This precision-guided approach promises to refine combination therapies, such as pairing immunotherapies with chemotherapies or targeted radiotherapy, or to repurpose existing drugs against novel genetic vulnerabilities uncovered by genome-wide CRISPR/Cas9 screens. Embedding emerging technologies such as single-cell sequencing and spatial transcriptomics into these platforms will unveil intratumoural heterogeneity at unprecedented resolution, further tailoring interventions to each patient’s unique tumour environment.

Realising this vision requires overcoming practical and conceptual challenges such as standardising co-culture protocols for reproducibility, scaling organoid and biobanks to capture CRC diversity, and integrating multi-omics datasets to draw actionable insights. Progress hinges on interdisciplinary collaboration that unites researchers, bioengineers, computational scientists, and clinicians to co-design experiments that reflect clinical realities. As these collaborative networks expand, the synergy of this pipeline will accelerate, driving a new era of rational, patient-centric therapies. Ultimately, this holistic paradigm moves us closer to the goal of predicting, preventing, and ultimately curing colorectal cancer.

## Figures and Tables

**Figure 1 cancers-17-03929-f001:**
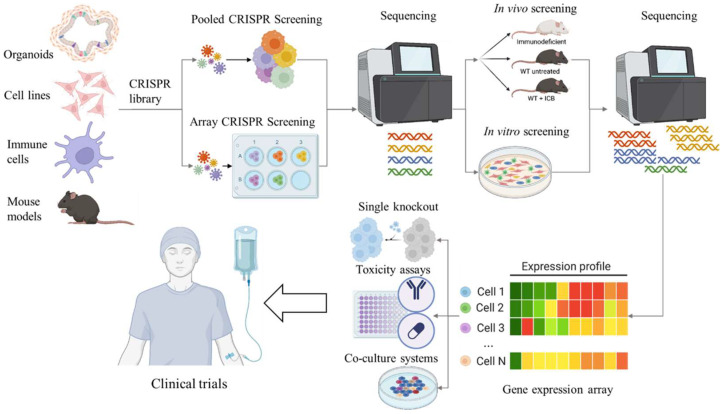
Schematic representation of the CRISPR/Cas9 workflow in the discovery and validation of novel immunotherapy targets for CRC. CRISPR screens can and have been performed on cell lines, organoids, immune cell populations, and mouse models. Initial CRISPR screens can be performed using pooled or arrayed libraries. Baseline sequencing is performed prior to immune pressure which could be conducted in vivo using immunocompetent mice, treating with immunotherapies, or using a co-culture system. Sequencing of the tumours or cells after intervention provides a list of potential target genes. The genes can be validated using conventional molecular techniques. Targeted therapies can then be established and tested in a clinical environment.

**Table 1 cancers-17-03929-t001:** Summary of the CRISPR screens undertaken in CRC systems.

Species	Model System	Library	Selection Pressure	Targets Identified	Validation Method	Year [Ref]
Mouse	Cell line-MC38/gp100	Genome-wide CRISPR library v2	T-cell mediated cytoxicity	PRMT1, RIPK1	Specific KO, inhibitor studies	2020 [64]
	Cell lines-MC38, CT26	Toronto Knockout Library	T-cell mediated cytoxicity, IFNγ	FITM2	Validation CRISPR screen (mVal)	2020 [65]
	Cell line-MC38	Commercial library	In vivo-anti-PD-1 treatment	PHF8	Over-expression and KD studies	2025 [66]
Human	Cell lines-HCT116, DLD1	Genome-wide LentiCRISPR v2	Drugs-Oxaliplatin, Irinotecan	PLK4	Targeted CRISPR screen	2024 [67]
	Cell lines-DLD1, RKO	Brunello	Cell growth	KMT2A	Epigenetic screen (EpiCK)	2021 [68]
	Cell line-HEK293	CRISPR KO library (GeCKO v2)	Cell growth	DHX29	Specific KO, targeted screen	2021 [69]
	Cell line-LoVo ERN1 KO cells	CRISPR KO library (GeCKO v2)	Drugs-Selumetinib, trametinib	DET1, COP1	Specific KO, inhibitor studies	2018 [70]
	Cell lines-SW620, HCT116	CRISPR KO library (GeCKO v2)	ABT-263 Bcl-XL Inhibitor	BCL-XL	Specific KO	2021 [71]
	Cell lines-HCT116, DLD1	CRISPR KO library (GeCKO v2)	In vivo-ThioNa or KHK inhibitor	INO80C	Targeted CRISPR screen	2017 [72]
	Cell lines-HCT116, SW480	CRISPR KO library (GeCKO v2)	AZD6244 inhibitor	GRB7	Specific KO, inhibitor studies	2021 [73]
	Cell lines-HT29, SW620	Commercial library	Cell growth	RAB10	Specific KO	2023 [74]
	Cell lines-HT29, HCT15, SW620	CRISPR KO library (GeCKO v2), Brunello	Primary NK cells	HLA-E, NCR3LG1	In vivo model	2021 [75]
	Cell line-LoVo	CRISPR KO library (GeCKO v2)	Intrasplenic injection	ANKRD42	Specific KO, in vivo model	2024 [76]
	Multiple cell lines	Human CRISPR library V1.0 and V1.1	Cell growth	WRN	Specific KO	2019 [77]
	Patient-derived cells	In house epigenetic library	Co-culture with PBMCs	EHMT2	In vivo model	2024 [78]
	Cell lines-POP92, POP66	In house epidrug library	Cell growth	HMGCR	Specific KO, inhibitor	2021 [79]
	Patient-derived organoids	Tumour suppressor gene (TSG) library	In vivo	TGFBR2	Validation screen	2020 [56]
	Patient-derived organoids	Brunello	TGF-β Selection	SWI/SNF complex	In vivo, TSG library screen	2020 [57]
	Patient-derived organoids	Human CRISPR library V1.0 and V1.1	Co-culture-anti-PD-1 treatment	WRN	In vivo PDX models	2021 [62]

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
