# Peer review of "Developing New Immunotherapy Approaches for Colorectal Cancer"

_cancers, 2025, doi:10.3390/cancers17243929_

Round 1
Reviewer 1 Report
Comments and Suggestions for Authors
This review presents current strategies for immunotherapy of colorectal cancer. This topic is relevant and clearly articulated. It presents current knowledge of genome-wide CRISPR/Cas9 screening methods and in vivo/in vitro validation using organoid and mouse models, which are well suited for testing hypotheses about immunotherapy resistance and target identification. The review is consistent with the journal's scope, covering cancer biology and immunotherapy, and addresses an important issue in colorectal cancer treatment: limited sensitivity to existing immunotherapies. The review incorporates recent advances and provides relevant experimental and clinical data.
The article is well written and logically structured, but for a more comprehensive coverage of this topic, the section on the influence of the gut microbiome on the effectiveness of immunotherapy and the progression of colorectal cancer in subchapter 2.2. Cancer cell extrinsic factors that influence immunotherapy response could be expanded. New articles have been published (2025) on the gut microbiota in colorectal cancer and its impact on immunotherapy.
The article identifies the immune checkpoints PD-1, PD-L1, and CTLA-4, and could also include newer ones such as LAG-3, TIM-3, and TIGIT.
The review does not address the use of circulating tumor DNA, exosomes, or immune profiling to predict response to immunotherapy.
Author Response
Comment 1: The article is well written and logically structured, but for a more comprehensive coverage of this topic, the section on the influence of the gut microbiome on the effectiveness of immunotherapy and the progression of colorectal cancer in subchapter 2.2. Cancer cell extrinsic factors that influence immunotherapy response could be expanded. New articles have been published (2025) on the gut microbiota in colorectal cancer and its impact on immunotherapy.
Response 1: Added this paragraph at line 152 (all line reference are in relation to the original document) to include new findings on gut microbiota in colorectal cancers and their impact on immunotherapies.
Recent studies identified 4-hydroxybenzeneacetic acid (4-HPA) as a key immunomodulatory metabolite secreted by gut microbiota leading to enhanced colorectal cancer progression (Liao et al 2025). The gut microbiota in colorectal cancer is significantly altered, with reduced biodiversity and abundance of certain species. CXCL3 release stimulated by 4-HPA directly recruits polymorphonuclear-MDSCs which in turn attenuates CD8+ T cell infiltration and activity. In a mouse setting, gut microbiota was found to synergize with anti-PD1 therapy, where combining treatment with broad spectrum antibiotics diminished tumour control (Cao et al 2025). Certain bacterial species such as that of Akkermansia muciniphila and Faecalibacterium prausnitzii were found to induce transcriptomic differences in immune cells. These changes translated into significantly altered general immune cell groups but as well as specific T cell and myeloid subsets. The effects of gut microbiota in immune mediated control of tumours are complex and it is clear certain species can be beneficial while others detrimental.
Comment 2: The article identifies the immune checkpoints PD-1, PD-L1, and CTLA-4, and could also include newer ones such as LAG-3, TIM-3, and TIGIT.
Response 2: Added this paragraph at line 81 to mention these emerging immunotherapy targets.
In colorectal cancer, particularly microsatellite-stable (MSS) tumours that often fail to respond to conventional checkpoint inhibitors, T-cell immunoglobulin and mucin-domain containing-3 (TIM-3), lymphocyte activation gene-3 (LAG-3), and T-cell immunoreceptor with Ig and ITIM domains (TIGIT) have gained attention as novel inhibitory receptors. These molecules have been shown to contribute to T-cell exhaustion and immune evasion, dampening anti-tumour immunity within the TME (Cai et al 2023, Mirestean et al 2025). Studies suggest that blocking TIM-3 can restore T-cell function, while LAG-3 inhibition synergizes with PD-1 blockade to enhance anti-tumour responses (Acharya et al 2020). TIGIT, which interacts with CD155 (also known as the poliovirus receptor (PVR)), suppresses T-cell and NK cell activity, and antibodies such as tiragolumab are being tested to counteract this pathway (Ghasemi 2025). Collectively, these checkpoints represent next-generation immunotherapy targets that may expand treatment options for colorectal cancer patients who do not benefit from current immunotherapies.
Comment 3: The review does not address the use of circulating tumor DNA, exosomes, or immune profiling to predict response to immunotherapy.
Response 3: I do like the use of circulating tumour DNA as a predictive tool for immunotherapy response – During my work with Melanoma, I discovered that the levels of MAP1LC3B found on circulating tumour DNA in metastatic melanoma patients could accurately predict immunotherapy response (Kelly et al 2021). While I did include a brief description of immune profiling, termed immunoscore – line 76, as a biomarker to predict response, I did not include any other methods. To fix this problem, I have added the following paragraph at line 78.
While these are not widely used in CRC patients, due to the limited use of immunotherapies, circulating tumour DNA (ctDNA) and exosomes are increasingly being recognized as valuable liquid biopsy tools for monitoring response to immunotherapy in cancer patients (Bartolomucci et al 2025, Niu et al 2025, Kumar et al 2025, Kelly et al 2021). ctDNA, released into the bloodstream from tumour cells, provides a dynamic snapshot of tumour burden and mutational landscape, with declining levels often correlating with treatment success and rising levels signaling early resistance or progression. Exosomes, which are small extracellular vesicles secreted by tumour and immune cells, carry proteins, RNA, and DNA that reflect both tumour biology and immune activity (Theodoraki et al 2019, Nie et al 2025). Together, ctDNA and exosomes offer minimally invasive biomarkers for predicting and detecting immunotherapy outcomes.
Reviewer 2 Report
Comments and Suggestions for Authors
Dear authors,
You’ve done a great and in-depth review of this challenging field showing the status and explaining the new ways of research. To my point of the are some points that need further clarifications.
- Lines 106 – 108: “As the tumour evolves, however, TGF-β can alter the tumour microenvironment and drive epithelial cells toward a mesenchymal, migratory state, that increases metastatic spread. ”This action of TGF-β should be explained and references must be shown.
- Lines 120 -124: “The ECM not only mechanically supports cancer tissues but also acts as a reservoir for growth factors, releasing them to cancer cells under specific conditions. It also acts as a scaffold that promotes cancer cell interactions with growth factors. Further- more, ECM re-modelling (via degradation) modulates the interaction of scaffold with can- cer cells and as well as regulates cellular metabolism and signalling pathways.” Here references are lacking.
- Lines 130 -131: “They can also express CD25 and convert ATP into Adenosine, restricting the availability of nutrients for T cells.” The ralation between the CD25 antigen and ATP metabolism should be explained.
- Line 133: Is the first mention of the acronym MDSC, so it meaning should be explained.
- Lines 140 – 142: “AMs are also capable of secreting angiogenic factors including VEGF, MCP-1, MIP-1a and MIP-2a, which promote infiltration of other immunosuppressive cells.” References are lacking.
Author Response
Comment 1: Lines 106 – 108: “As the tumour evolves, however, TGF-β can alter the tumour microenvironment and drive epithelial cells toward a mesenchymal, migratory state, that increases metastatic spread”. This action of TGF-β should be explained and references must be shown.
Response 1: I expanded on the role of TGF-β and added the following at line 107.
Within the tumour microenvironment, TGF-β fosters immunosuppression by inhibiting cytotoxic T cell and NK cell activity, while simultaneously enhancing regulatory T cell and myeloid-derived suppressor cell function. At the cellular level, TGF-β signaling induces epithelial-to-mesenchymal transition (EMT), a process in which epithelial cells lose polarity and adhesion, acquire mesenchymal traits, and gain migratory and invasive capabilities. This EMT not only facilitates metastatic spread but also contributes to therapy resistance by creating a more aggressive, stem-like phenotype. By reshaping both immune and stromal compartments, TGF-β acts as a central driver of tumour progression, making it a possible target for therapeutic strategies aimed at overcoming metastasis and improving immunotherapy efficacy (Shi et al 2022, Cecerska-Heryc et al 2025, Himani et al 2025).
Comment 2: Lines 120 -124: “The ECM not only mechanically supports cancer tissues but also acts as a reservoir for growth factors, releasing them to cancer cells under specific conditions. It also acts as a scaffold that promotes cancer cell interactions with growth factors. Further- more, ECM re-modelling (via degradation) modulates the interaction of scaffold with cancer cells and as well as regulates cellular metabolism and signalling pathways.” Here references are lacking.
Response 2: Added 2 references – Lee et al 2025 and Chitty et al 2025 – these were accidently removed during editing
Comment 3: Lines 130 -131: “They can also express CD25 and convert ATP into Adenosine, restricting the availability of nutrients for T cells.” The ralation between the CD25 antigen and ATP metabolism should be explained.
Response 3: Expanded the relationship between CD25 and ATP metabolism by adding the following sentences at Line 131.
CD25 acts as a gatekeeper for IL-2 signaling, directly shaping ATP metabolism in T cells. Effector T cells depend on CD25 to fuel glycolysis and expansion, while Tregs use constitutive CD25 expression to monopolize IL-2, sustaining oxidative phosphorylation-driven suppressive functions. This balance is central to immune regulation and a key target in immunotherapy (Brusko et al 2009, Seo et al 2024).
Comment 4: Line 133: Is the first mention of the acronym MDSC, so it meaning should be explained.
Response 4: Myeloid derived suppressor cells were first mentioned in line 125 as part of the immune milieu, I have expanded on what these cells are and their function by adding the following sentences at line 134.
with potent immunosuppressive activity that originate from the bone marrow's myeloid progenitors. They are involved in many pathological conditions, such as cancer, and work by suppressing the immune system's response, particularly T cells, and by promoting processes like tumour growth, angiogenesis, and metastasis. MDSCs are being studied as potential therapeutic targets in cancer treatment. (Wang et al 2020).
Comment 5: Lines 140 – 142: “AMs are also capable of secreting angiogenic factors including VEGF, MCP-1, MIP-1a and MIP-2a, which promote infiltration of other immunosuppressive cells.” References are lacking.
Response 5: Added reference – Basak et al 2023, Huang et al 2024.